# Serum Extracellular Vesicle-Derived hsa-miR-2277-3p and hsa-miR-6813-3p Are Potential Biomarkers for Major Depression: A Preliminary Study

**DOI:** 10.3390/ijms241813902

**Published:** 2023-09-09

**Authors:** Issei Seki, Hiroto Izumi, Naomichi Okamoto, Atsuko Ikenouchi, Yasuo Morimoto, Seichi Horie, Reiji Yoshimura

**Affiliations:** 1Department of Psychiatry, University of Occupational and Environmental Health, Kitakyusyu 807-8555, Japan; issei-seki@med.uoeh-u.ac.jp (I.S.); nokamoto@med.uoeh-u.ac.jp (N.O.); atsuko-i@med.uoeh-u.ac.jp (A.I.); 2Center for Stress-related Disease Control and Prevention, University of Occupational and Environmental Health, Kitakyusyu 807-8555, Japan; h-izumi@med.uoeh-u.ac.jp (H.I.); yasuom@med.uoeh-u.ac.jp (Y.M.);; 3Department of Occupational Pneumology, Institute of Industrial Ecological Sciences, University of Occupational and Environmental Health, Kitakyusyu 807-8555, Japan; 4Department of Health Policy and Management, Institute of Industrial Ecological Sciences, University of Occupational and Environmental Health, Kitakyusyu 807-8555, Japan

**Keywords:** major depression, extracellular vesicles, hsa-miR-6813-3p, hsa-miR-2277-3p, serum

## Abstract

The aim of the present study was to examine the association between miRNA levels in extracellular vesicles (EVs) from serum and the severity of Major Depression (MD). Patient sera from 16 MD cases were collected at our university hospital. The miRNAs contained in EVs were extracted using a nanofiltration method, and their expression levels were analyzed using miRNA microarrays. Intergroup comparisons were performed to validate the diagnostic performance of miRNAs in EVs. Furthermore, candidate miRNAs in EVs were added to neural progenitor cells, astrocytes, and microglial cells in vitro, and the predicted target genes of the candidate miRNAs were extracted. The predicted target genes underwent enrichment analysis. The expression levels of hsa-miR-6813-3p and hsa-miR-2277-3p were significantly downregulated with increasing depression severity of MD. The pathway enrichment analysis suggests that hsa-miR-6813-3p may be involved in glucocorticoid receptor and gamma-aminobutyric acid receptor signaling. Additionally, hsa-miR-2277-3p was found to be involved in the dopaminergic neural pathway. The analysis of serum miRNAs in EVs suggests that hsa-miR-6813-3p and hsa-miR-2277-3p could serve as novel biomarkers for MD, reflecting its severity. Moreover, these miRNAs in EVs could help understand MD pathophysiology.

## 1. Introduction

Major depression (MD) is one of the most disabling psychiatric disorders, with an estimated lifetime prevalence of ~2–21% [1]. Individuals with MD have a higher risk of suicide mortality compared to healthy individuals [2]. Assessing depression severity in MD is critical for treatment, as the efficacy of antidepressants and other treatments depends on their intensity [3]. The Hamilton Rating Scale for Depression (HAMD) is a standardized scale widely used to assess the severity of MD; however, HAMD scores are not well matched across observers. The development of biomarkers that can more rigorously assess disease diagnosis and severity is an important challenge.

Useful biomarkers for the severity of MD have not yet been identified. However, several candidates have been proposed as biomarkers for MD. These include inflammatory cytokines [4,5], brain-derived neurotrophic factor (BDNF) [6,7], oxidative stress markers [8,9], and neuroimaging [10]. However, no biomarker has been found to have sufficient sensitivity or specificity. Recently, miRNAs have attracted attention as diagnostic biomarkers of MD [11,12].

miRNAs are small noncoding RNA (ncRNA) molecules formed by an average of 22 nucleotides. They bind to the 3′ untranslated region (UTR) of target messenger RNAs (mRNAs), causing translational repression and consequent suppression of the target mRNA expression [13]. miRNAs have been reported to exhibit variable expression levels in diabetes mellitus, cardiovascular diseases, cancer, and infectious diseases. Moreover, MiRNAs have been implicated in psychiatric and neurodegenerative disorders [14].

It has been suggested that miRNAs are involved in the regulatory mechanisms of gene expression and are deeply involved in the neurobiological basis of psychiatric diseases, including MD [15,16] and schizophrenia [17]. Recent studies have indicated that miRNAs play important roles in neuroplasticity, neurogenesis, and stress response [16,18].

Extracellular vesicles (EVs) are a general term encompassing membrane vesicles secreted by cells, which include well-known examples such as exosomes and macrovesicles. Exosomes are vesicles with a 40–100 nm lipid bilayer structure formed by endocytosis and secreted by numerous cells [19]. Exosomes have been reported to be widely distributed in human body fluids, such as saliva, blood, breast milk, amniotic fluid, and urine [20,21,22,23]. The lipid duplex structure of exosomes protects miRNAs, which allows them to circulate stably outside cells. miRNAs are transported via exosomes to other cells, where they are thought to regulate the biological activity of recipient cells [24,25].

Thus, EVs containing exosomes are stable extracellularly and are being extensively investigated as potential new biomarkers for MD. In the present study, we examined the expression profiles of miRNAs in EVs present in the blood with the aim of identifying biomarkers that correlate with the severity of MD. Additionally, we performed an in silico analysis to identify miRNAs targets that have been validated in vitro by other research groups.

## 2. Results

### 2.1. Patient Characteristics

This study enrolled 16 patients diagnosed with MD, aged 34 to 63 years (Table 1). The mean age of the MD group was 50.13 years, and 50% of the participants were women, with a mean HAMD score of 16.50. Three participants had mild depression (median age, 53.0 years; HAMD, 12.0 points), ten had moderate depression (median age, 48.5 years; HAMD, 16.0 points), and three had severe depression (median age, 49.0 years; HAMD, 23.0 points). There were no significant differences in age (*p* = 0.555) or sex (*p* = 0.354) between the mild, moderate, and severe depression groups (Table 2). Twelve of the sixteen participants were taking antidepressants and ten were taking benzodiazepines (Table 1). For the healthy control group, a total of 284 individuals were enrolled. There were 209 males (87.08%) and 31 females (12.92%). The age distribution in the control group showed an average of 32.89 years with a standard deviation of 10.36 years. The age range spanned from 19 to 67 years, with the first quartile at 24 years, the median at 32 years, and the third quartile at 39 years.

### 2.2. hsa-miR-6813-3p, hsa-miR-2277-3p, and hsa-let-7f-1-3p in EVs Expression Decreases with Increasing Severity of Depression

In this study, EVs were isolated from the serum and cell culture medium by nanofiltration. EVs isolated by this method were spherical, as established by transmission electron microscopy (TEM), and the EV diameter was ~170 nm, as determined by dynamic light scattering (DLS). EVs contained CD9, CD63, CD81, and EpCAM proteins, as seen by Western blotting, and contained miRNAs less than 30 bp in length as revealed by an Agilent 2100 Bioanalyzer [26].

When miRNA expression levels were analyzed in 16 cases of MD, significant differences were found in 28 miRNAs, as shown in Appendix A (*p* < 0.05). When comparing between groups in terms of severity by the HAMD score, 25 miRNAs showed no change in severity. hsa-miR-6813-3p, hsa-miR-2277-3p, and hsa-let-7f-1-3p were significantly downregulated with increasing severity, based on HAMD scores (Appendix A). In addition, to distinguish MD patients from healthy controls, we also performed a subgroup analysis of the three compared miRNAs. The results of this analysis, in comparison to healthy controls, are shown in both Figure 1 and Table 3; we found that hsa-miR-6813-3p and hsa-miR-2277-3p were significantly downregulated in MD patients. hsa-miR-6813-3p and hsa-miR-2277-3p decreased with increasing severity of the HAMD score and showed greater statistical significance (*p* < 0.01) in Mann–Whitney U and Kruskal–Wallis tests. Importantly, these *p*-values remained significant even after FDR correction. However, hsa-let-7f-1-3p showed no significant differences. These results suggest that low expression of hsa-miR-6813-3p and hsa-miR-2277-3p is significantly associated with MD severity.

Further analysis revealed a significant and strong negative correlation between the expression levels of hsa-miR-6813-3p (Spearman’s rank correlation coefficient = −0.62, *p* = 0.0104) and hsa-miR-2277-3p (Spearman’s rank correlation coefficient = −0.671, *p* = 0.00446) and the severity of depression as measured by the HAMD score (Figure 2). This suggests that as depression severity increases, the expression of these miRNAs decreases.

The values in Table 3 represent miRNA signal values adjusted by subtracting the mean of the Negative Control ± 2SD. MiRNAs with non-positive adjusted values are shown as “0”. Valid miRNAs have their adjusted signal values displayed.

Each cell value represents the median expression level or statistical *p*-value of the corresponding miRNA.

The “2-Group *p*-value” indicates the result of the Man–Whitney test between the MDD and Control groups.

The “4-Group *p*-value” indicates the result of the Kruskal–Wallis test among the Control, Mild, Moderate, and Severe groups.

The “2-Group FDR *p*-value” and “4-Group FDR *p*-value” represent the False Discovery Rate (FDR) corrected *p*-values for the respective comparisons between MDD and Control, and among Control, Mild, Moderate, and Severe groups.

FDR correction was performed using the Response Screening feature in JMPpro, based on one-way analysis of variance (ANOVA).

The scatter plots represent the relationship between the severity of depression (*x*-axis), HAMD score severity, and miRNA expression levels (*y*-axis).

### 2.3. Target Gene Prediction of hsa-miR-6813-3p and hsa-miR-2277-3p

We found that hsa-miR-6813-3p and hsa-miR-2277-3p in serum EVs were significantly decreased in MD patients. Therefore, we forced neuronal precursor cells, microglia, and astrocytes to express these miRNAs and searched for genes whose expression decreased. The reasons for the forced expression of miRNAs in cultured cells was that these three cell lines had low levels of miRNA expression (Table 3); we hypothesized that these cells take up miRNAs from EVs in serum and maintain their neural functions. In other words, we thought that the expression of genes that suppress nerve functions could be reduced by the uptake of miRNAs.

Candidate target genes for each cell with decreased expression compared to the control cells were extracted, and their assembly relationships are shown in a Venn diagram (Figure 3). Astrocytes showed the most expression changes compared to neural progenitor cells and microglia. Since astrocytes have been suggested to be strongly involved in the action of two types of miRNAs, hsa-miR-6813-3p and hsa-miR-2277-3p, astrocytes were used for target gene prediction. A list of these downregulated genes is provided in Appendix A. In addition, a comprehensive list of genes, including their specific expression status (upregulated or downregulated) along with their expression values, is provided in Appendix A. Astrocytes were found to target 215 genes for hsa-miR-6813-3p and 2047 genes for hsa-miR-2277-3p. miRNA target genes were searched using TargetScan TM in the miRNA database and compared with the in vitro predicted miRNA target genes. Specifically, genes affected by these miRNAs and their overlap with TargetScan TM predictions are detailed in Appendix A. Hsa-miR-6813-3p target genes included BEND4 and SERPING1. Hsa-miR-2277-3p target genes included STN2, BEND4, HLA-DQB1, KYNU, PCDHA4, PCDHA6, and ZSCAN12.

### 2.4. GO Analysis and Pathway Enrichment Analysis of hsa-miR-6813-3p and hsa-miR-2277-3p

GO analysis was performed to identify the biological processes involved in the identified candidate target genes. As shown in Figure 4, the results of the GO analysis listed those with *p* < 0.01 and the top 20 functions. hsa-miR-6813-3p is mainly located in the extracellular matrix of cellular components (CCs) and is involved in biological processes (BPs) such as skeletal system development, the enzyme-linked receptor protein signaling pathway, and lymphoid progenitor cell differentiation. Hsa-miR-2277-3p is mainly located in the extracellular matrix and axons of CCs, which are involved in BPs such as muscle tissue development and cellular component morphogenesis, and molecular functions (MFs) (e.g., inorganic molecular entity membrane transporter activity and signaling receptor regulator activity). In the pathway enrichment analysis by Wikipathways, the results of the pathway analysis associated with the predicted target genes of hsa-miR-6813-3p showed the glial cell line-derived neurotrophic factor/rearranged during transfection signaling axis, oxidative damage, estrogen receptor pathway, glucocorticoid receptor pathway, and gamma-aminobutyric acid (GABA) receptor signaling (Table 4). The pathway analysis results from the predicted target genes of hsa-miR-2277-3p confirmed the involvement of multiple processes in the pathophysiology of MD, including dopaminergic neurogenesis, thyroxine (thyroid hormone) production, oxidative stress, oxidative stress response, tryptic production, tryptophan metabolism, nitric oxide-cyclic guanosine monophosphate-protein kinase G pathway-mediated neuroprotection, and methylation pathways (Table 5).

## 3. Discussion

In the present study, EVs were isolated from the blood samples of both patients with MD and healthy controls. The expression levels of miRNAs in EVs were comprehensively analyzed by microarray analysis to identify two biomarkers (hsa-miR-6813-3p and hsa-miR-2277-3p) that could noninvasively diagnose the severity of MD.

Although our study is among the first to associate EV-miRNAs with the severity of depression, there are other studies in the realm of neuropsychiatric disorders that have used EVs as a source of biomarkers. Wei et al. [27] examined the miRNA expression profile of serum exosomes in MD patients and identified hsa-miR-139-5p as being crucial in the pathogenesis of MD. Another study by Ran et al. [28] highlighted the impact of childhood trauma on serum EV microRNA imbalances in adolescents with MD. Additionally, Honorato-Mauer et al. [29] investigated alterations in EV microRNA related to various neuropsychiatric disorders, including MD, in adolescents. Recently, the relevance of EVs, particularly in relation to the central nervous system and as potential non-invasive diagnostic markers for various diseases, has been highlighted by Oraki Kohshour et al. [30]. Moreover, several previous studies on miRNAs in MD utilized a range of sample types, including postmortem brain tissue [31], cerebrospinal fluid [32], whole blood [33], serum [34], and plasma [35], as samples and reported various miRNA expression patterns. These studies suggest that under pathophysiological conditions, neurons release EVs containing different miRNAs and transmit specific information. Despite the emerging evidence linking EV-miRNAs with neuropsychiatric disorders, there seems to be a lack of reproducibility in the field. In a comprehensive review of the literature on miRNAs associated with MD [36], very few miRNAs were consistently expressed across studies, suggesting that a source of discrepancy in miRNA expression may be methodological differences, such as extraction methods. Therefore, the extraction of EVs and exosomes has attracted attention because miRNAs are stable in exosomes in the EVs of body fluids. Several methods have traditionally been employed to separate exosomes, including differential and suspended density centrifugation, ultrafiltration, size exclusion, precipitation, and immunoaffinity separation [37]. As indicated by the comprehensive review by Théry et al. [38], there is still no consensus on the optimal method for EV including exosomes isolation. Their findings suggest that although high recovery and specificity are desired, they might be unachievable with the current methods. In our study, we employed the nanofiltration method for EVs containing exosomes and miRNAs isolation, which is different from some previous studies that used centrifugation. Since the nanofiltration method captures particles depending on their size, it has the advantage of being able to capture many extracellular vesicles, including exosomes, in a short time and at low cost. However, since the sizes of exosomes and microvesicles partially overlap, there is the drawback that the next analysis must be performed while these are mixed [39]. Previously, it was thought that tetraspine (CD9, CD63 etc.) was specifically expressed on the exosome membrane and could be used to enrich exosomes. However, since it was found that some tetraspanins (CD81, CD63) are also expressed in the membrane of microvesicles [40], there is currently no method for isolating exosomes alone. The difference between the miRNAs we identified and previous reports is likely due to the amount of microvesicles contamination, due to the isolation method. Therefore, we believe that nanofiltration can be used as a means of separating extracellular vesicles.

EVs and exosomes can cross the blood–brain barrier (BBB) and may serve as biomarkers of mental disorders [41]. Although the relevant mechanisms are not yet fully understood, there is evidence that exosomes can pass through the BBB [42]. One potential mechanism involves the incorporation of exosomes into the BBB and their transportation across the barrier via transcytosis. Another possibility is that exosomes interact with immune cells and induce them to transport vesicles across the barrier. EVs released from brain cells can also be extracted from blood [43]. Exosomes are small vesicles secreted by various cells and contain a variety of biomolecules, including proteins, lipids, and nucleic acids. These vesicles play an important role in intercellular communication, and they have been attracting attention for various potential treatments for CNS disorders. Overall, the ability of exosomes to traverse the BBB may represent a promising approach for the delivery of therapeutics to the CNS.

There were two main findings regarding the roles of hsa-miR-6813-3p and hsa-miR-2277-3p in MD. However, it is challenging in this study to distinguish between the potential effects of MD on these miRNAs and the possibility that these changes in signaling may be influenced by the use of antidepressants or other reported drugs. Nevertheless, the significant differences observed in the expression levels of these miRNAs between healthy participants and MD patients suggests that these changes may reflect the underlying pathophysiology of MD.

First, we compared the expression levels of extracellular vesicle miRNAs in MD by microarray analysis and identified three types of miRNAs whose expression levels were significantly reduced with depression severity: hsa-miR-6813-3p, hsa-miR-2277-3p, and hsa-let-7f-1-3p, each of which were significantly downregulated according to the severity of MD. Our study also compared these three miRNAs with those of healthy controls to distinguish patients with MD from healthy participants. The expression of hsa-miR-6813-3p and hsa-miR-2277-3p was significantly lower in the MD group than in the healthy control group. Previous work has reported that miR-6813-3p is involved in ASD [44]. Prior research has also shown that miR-2277-3p is involved in colon cancer cells [45]. To the best of our knowledge, this study is the first to describe a significant decrease in the expression levels of hsa-miR-6813-3p and hsa-miR-2277-3p with increasing severity of MD. 

Second, we searched for candidate target genes involving hsa-miR-6813-3p and hsa-miR-2277-3p in vitro and performed GO and pathway enrichment analyses. Our decision to overexpress the identified miRNAs, hsa-miR-6813-3p and hsa-miR-227-3p, in specific brain cell types stemmed from several key observations. Firstly, a marked reduction of these miRNAs was noted in the serum EVs of patients with depression compared to healthy individuals. Additionally, in the cells we utilized for our experiments, the expression levels of these two miRNAs were notably low. Based on these findings, we hypothesized that neurons, despite their low intrinsic miRNA expression, might acquire these miRNAs from serum EVs, which could be instrumental in maintaining their normal function. Supporting our hypothesis is a foundation of prior studies that have demonstrated the capability of extracellular vesicles, specifically exosomes, to traverse the blood–brain barrier bidirectionally. Notably, the study presented by [46] provides an in-depth examination of the mechanism through which exosomes migrate into brain tissues. Furthermore, research indicated in [47,48,49,50] reports observations of neuroregenerative effects when exosomes are intravenously injected into mice. Also, in the study conducted by Chen et al. [51], it was confirmed that exosomes derived from mesenchymal stem cells (MSC) enhance the function of astrocytes in an Alzheimer’s disease model. To test this hypothesis, we introduced these miRNAs into cultured cells with low miRNA expression, aiming to decipher the positive mechanisms exerted on neural function. During this process, our research primarily focused on identifying the genes potentially impacted by these miRNAs. Enhancing the activity of these miRNAs allowed us to identify the associated target genes more accurately. In the extraction of the predicted target genes by microarray, hsa-miR-6813-3p and hsa-miR-2277-3p showed the greatest changes in expression in astrocytes. Astrocytes have been previously associated with depression-like behavior in mice and postmortem brain tissues [52]. hsa-miR-6813-3p and hsa-miR-2277-3p exert regulatory effects on astrocytes, suggesting that they may play important roles in the regulation of biological functions. Therefore, we also focused on astrocytes. Predicted target genes extracted in vitro were compared to the target genes in the target scan. The candidate target genes for hsa-miR-6813-3p were BEND4 and SERPING1, and those of hsa-miR-2277-3p included STN2, BEND4, HLA-DQB1, KYNU, PCDHA4, PCDHA6, and ZSCAN12. These genes have been previously associated with depression [53].

In the GO process obtained from GO analysis by Metascape, biological processes associated with lymphoid progenitor cell differentiation were revealed in hsa-miR-6813-3p. Previous studies have reported an association between MD, stress, and the modulation of the immune system [54], which may be partly due to the altered expression of hsa-miR-6813-3p. hsa-miR-2277-3p, interestingly, shows a strong association with brain neurotransmitters, including axon cellular components, and these functions of hsa-miR-2277-3p may be involved in the pathogenesis of MD.

The pathway enrichment analysis suggested that hsa-miR-6813-3p may be involved in the glucocorticoid receptor pathway and GABA receptor signaling. In addition, hsa-miR-2277-3p may effect dopaminergic neurogenesis and dopaminergic neuronal pathways. Several brain neurotransmitter systems, including glutamate, γ-aminobutyric acid, serotonin, norepinephrine, and dopamine, are implicated in MD as well as bipolar disorder [55]. We hypothesized that the impairment of these processes might influence the severity of MD.

Our study has several limitations that should be acknowledged. Firstly, the sample size was small, and the number of patients with MD was highly skewed, with only three cases each classified as mild and severe. Secondly, the study design was cross-sectional. Thirdly, the administration of antidepressants and benzodiazepines to the participants may have influenced the miRNA expression levels. Fourthly, the validation of microarray data alone is not sufficient to rule out false-positive results. Fifthly, we observed differences in miRNA expression between men and women, but we were unable to conduct multivariate analysis incorporating covariates such as age, gender, drug type, and medication dosage. Therefore, the results should be interpreted with caution. In summary, the classification of MD severity in our study relied solely on the HAMD score, and the methodological validity of comparing miRNA levels between groups as a biomarker requires further comprehensive investigation. Future research efforts should involve cohort studies with larger sample sizes or drug-naïve patients with MD using a restrict design to confirm the preliminary results of this study.

## 4. Materials and Methods

### 4.1. Subjects

All participants with MD were recruited as outpatients or inpatients at the Occupational and Medical University Hospital, Japan between September 2018 and March 2020. Patients with MD were outpatients or inpatients between the ages of 20 and 65 years and met the diagnostic criteria for MD according to the Diagnostic and Statistical Manual of Mental Disorders (5th edition, DSM–5, American Psychiatric Association, 2013) [56].

A healthy control group comprising 284 individuals aged 20–68 years, with no history of diabetes, cardiovascular disease, malignant tumor, infectious disease, or psychiatric disease was included. This sample was collected for a future study yet to be conducted. It is important to note that all participants were of Japanese descent. The control group comprised individuals employed by a particular entity, the identity of which has been withheld to maintain confidentiality. It should be noted that specific demographic data, such as smoking habits, alcohol consumption, and educational background, were not available for this analysis. Participants in the control group were either full-time employees or associates affiliated with the said entity.

This study was approved by the Ethics Review Committee of the University of Occupational and Environmental Health Sciences in accordance with the Declaration of Helsinki (UOEHCRB21-083). All participants provided written informed consent. The analysis of miRNA expression levels in the healthy control group was performed with approval (H30-021) from the Ethics Review Committee of the University of Occupational and Environmental Health.

### 4.2. Psychometric Evaluation

One experienced psychiatrist evaluated the 17-item HAMD [57] for depressive symptoms in all participants. The severity of depression was classified using the HAMD scores as follows: mild (8–13), moderate (14–18), and severe or greater (≥19) [58].

### 4.3. Laboratory Method

Ten mL of blood was collected in blood collection tubes from patients diagnosed with MD and 2 mL of blood from healthy participants. Blood samples were allowed to clot at room temperature by rotating slowly for over 5 min. After confirming the coagulation of the blood, it was stored at 4 °C and the following procedure were performed within 4 days. After centrifuging the tubes at 3000 rpm for 10 min, supernatant serum was stored at −80 °C until analysis.

### 4.4. Isolation of EVs from Serum and Purification of miRNA in EVs

Solubilized serum (0.5 mL) was diluted with 1 mL saline, and the whole volume (1.5 mL) was aspirated into a 2.5 mL syringe. Syringe filters of 220 nm (SFPES013022N; Membrane Solutions, Auburn, WA, USA) and 50 nm (SF16008; TISCH Scientific, Cleves, OH, USA) were connected to the syringe in that order. The plunger was pushed at a speed of one drop per second or less to pass through the syringe filter. Then, 2 mL of saline was aspirated into a 2.5 mL-syringe and attached to a connected syringe filter. The syringe filter was washed by pressing the plunger at a speed of one drop per second or less. One milliliter of ISOGEN (Nippongene, Ajman, United Arab Emirates) was aspirated into a 1 mL-syringe and connected to a washed 50 nm-syringe filter. The plunger was moved up and down to remove air from the syringe filter, and the sample was recovered for 1 min. The tube containing the sample was vortexed for 1 min. Chloroform (200 μL) was added to the tube, mixed using a vortex mixer for 15 s, and left at room temperature for 2–3 min. After centrifugation at 12,000× *g* for 15 min, 450 µL of the supernatant and 450 µL of 2-propanol were mixed using a vortex mixer. The whole volume (900 μL) was passed through the NucleoSpin-miRNA-Plasma kit (MACHEREY-NAGEL, Düren, Germany). Next, miRNA in the EVs was immobilized on the column according to the NucleoSpin-miRNA-Plasma kit protocol. miRNA was extracted by passing 50 μL of H2O through the column twice (total volume of 100 µL). Ten microliters of 3 M potassium acetate (pH 5.5), 100 µL of 2-propanol, and 1 µL of pellet paint were added to 100 μL of miRNA extracted tubes and centrifuged at 21,000× *g* (15,000 rpm) for 15 min at 4 °C. After washing the pellet with 250 μL of 70% ethanol, it was air-dried and stored at −80 °C until microarray analysis.

### 4.5. Formation of Cells Stably Expressing miRNA and Purification of Total RNA

Immortalized human dopaminergic neuronal precursor cell line (LUHMES cells, Cat. No. T0284), immortalized human microglia (Cat. No. T0251), and immortalized human astrocytes (Cat. No. T0280) were obtained from Applied Biological Materials Inc. (Richmond, BC, Canada) and COS1 (Cat. No. CRL-1650) was purchased from the American Type Culture Collection (Manassas, VA, USA). Cells stably expressing miRNAs were established using lentivirus with a biological replicate of 1. Briefly, pSIH1-H1-copGFP-T2A-Puromycin was constructed using pSIH1-H1-copGFP (SI501A-1, SBI: System Biosciences, Palo Alto, CA, USA). Oligonucleotides synthesizing hsa-miR-2277-3p and hsa-miR-6813-3p (Appendix A) were inserted into this plasmid at the BamHI-EcoRI site, downstream of the H1 promoter. A vector without insert was used as a control. The virus was produced by COS1 cells using a pPACKH1 Lentivector HIV Packaging Kit (LV500A-1, SBI) for virus packaging. The virus-containing cell culture medium was passed through a 0.45 μm syringe filter and added to the culture medium of neural precursor cells, astrocytes, and macroglia. Cells expressing miRNAs were selected by culturing with puromycin for more than two weeks, and 100% of the cells were confirmed to express green fluorescent protein (GFP). To verify the expression of hsa-miR-6813-3p and hsa-miR-227-3p, additional RT-qPCR experiments were conducted. It was confirmed that miRNAs were overexpressed in cells transfected with hsa-miR-6813-3p or hsa-miR-227-3p (Appendix A). Total RNA was purified using the miRNeasy Mini Kit (Qiagen, Hilden, Germany). Purified total RNA was confirmed to be of high purity using a 2100 Bioanalyzer System (Agilent, Santa Clara, CA, USA).

### 4.6. Microarray Analysis

Microarray analysis of miRNA was performed as previously described [26] using a Human miRNA Oligo Chip 4 plex (TORAY, Tokyo, Japan) containing 2565 probe sets. For the miRNA array analysis in the control group, a total of 300 sera samples were collected. Each of these 300 samples was analyzed individually, without pooling. Of these, 284 samples were retained for the final dataset, with the remaining 16 samples excluded due to defects found either in the samples themselves or in the array chips. Microarray analysis of mRNA was similarly performed as described in [59], utilizing a Human Oligo Chip 25k set (TORAY) containing 24,460 probe sets. Each miRNA and mRNA was normalized using the global normalization method, which adjusted the median of the detected signal intensity to 25.

### 4.7. Reverse Transcription and Quantitative Real-Time PCR Analysis (RT-qPCR)

Total RNA including miRNA was purified with an miRNeasy Kit (217084; Qiagen, Venlo, The Netherlands) from neuronal precursor cells, microglia, and astrocytes stably transfected with hsa-miR-2277-3p, hsa-miR-6813-3p, and empty miRNA. RT-qPCR assay was then performed according to the manufacturer’s instructions, for which the primer sets were as follows: 479594_mir for hsa-miR-2277-3p, 480391_mir for hsa-miR-6813-3p, and 001973 for U6 snRNA (Applied Biosystems, Foster City, CA, USA). Values were normalized to those of U6 snRNA. The comparative cycle time method was used to quantify gene expression. All samples were analyzed in single in each experiment.

### 4.8. Selection of Candidate Target Genes

Data analysis of the normalized microarray was performed using Gene Spring GX (Agilent Technologies, Inc., Santa Clara, CA, USA). Neural progenitor cells, astrocytes, and microglia were compared to control cells; expression was negatively regulated, and those with a 2-fold or greater difference in expression were designated as candidate target genes. For a comprehensive understanding of miRNA-target interactions, we utilized the TargetScan™ database. Inputting the miRNAs hsa-miR-6813-3p and hsa-miR-2277-3p, we extracted predicted target genes. These predicted targets were then cross-referenced with our microarray findings to identify overlaps and potential discrepancies.

### 4.9. GO and Pathway Enrichment Analysis

To predict the potential function of candidate target genes obtained from experimental mRNA array analysis, Metascape (https://metascape.org/) (accessed on 22 November 2022) was used to annotate each gene based on Gene Ontology (GO) and to perform pathway enrichment analysis. Metascape is a web-based tool designed to allow experimenters to apply a powerful computational analysis pipeline to analyze and interpret large datasets [60]. GO results were statistically significant at a threshold of *p* < 0.01, and the top 20 were ranked based on the −log10 (*p* value) of the analysis by Metascape. In Metascape, FDR correction was conducted using the Benjamini-Hochberg method, and adjusted *p*-values were employed [60]. Pathway analysis using Wikipathways was performed with Gene Spring GX. The pathway analysis was considered statistically significant at a threshold of *p* < 0.05.

### 4.10. Statistics

Statistical analyses were performed using EZR (Saitama Medical Center, Jichi Medical University, Vienna, Austria), which is a graphical user interface for R (R Foundation for Statistical Computing, Saitama, Japan). Moreover, a modified version of R Commander was designed to add the statistical functions frequently used in biostatistics [61]. Normal distribution data is presented as mean ± standard deviation, non-normal distribution data is presented as median (quartile deviation), and categorical variables are presented as counts (percentages). The chi-square test was used to compare categorical data, and the Kruskal–Wallis test was used for comparisons among the three groups. The Kruskal–Wallis test was applied for group comparisons of miRNA expression levels because normality in the severity of depression was not found in comparisons between the three groups. The Mann–Whitney U test was used to compare the control and depression groups. Statistical significance was determined at a threshold of *p* < 0.05. It is important to note that only univariate analysis was performed in this study. Multivariate analysis with adjustment for covariates including sex, age, and medication dosage could not be conducted due to the small sample size of the MD group. We applied a False Discovery Rate (FDR) correction to the *p*-values obtained from our statistical tests to account for the issue of multiple comparisons. This FDR correction was performed using the Response Screening feature in JMPpro, which is based on one-way analysis of variance (ANOVA).

## 5. Conclusions

The expression levels of hsa-miR-2277-3p and hsa-miR-6813-3p in EVs decreased with the increased severity of MD. Thus, hsa-miR-2277-3p and hsa-miR-6813-3p may be potential biomarkers of MD severity and may also be associated with its severity. We should be cautious in our interpretation of these results, as it is difficult in this study to distinguish between the potential effects of MD on these miRNAs and the possibility that the use of antidepressants or other reported medications may have caused changes in this signaling.

## Figures and Tables

**Figure 1 ijms-24-13902-f001:**
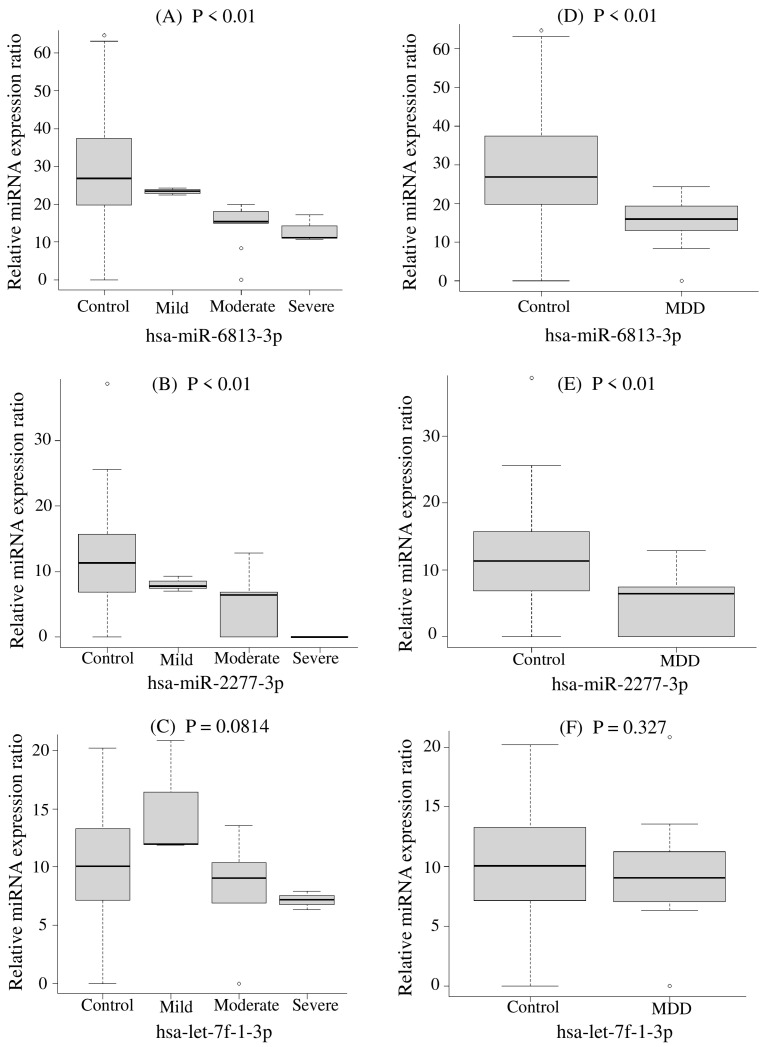
Subgroup analysis by comparison of the healthy and major depression groups. The Kruskal–Wallis test was performed for (**A**–**C**). Mann–Whitney’s U test was performed for (**D**–**F**).

**Figure 2 ijms-24-13902-f002:**
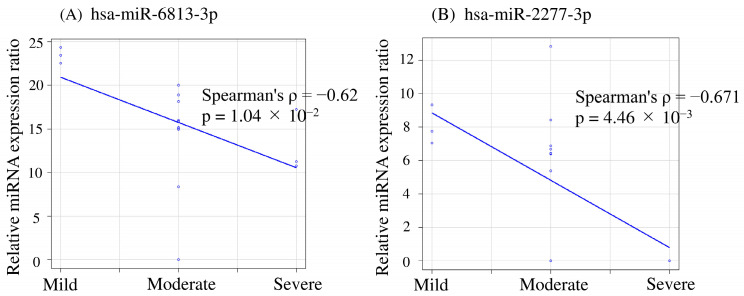
Spearman’s correlation between severity of depression and miRNA expression.

**Figure 3 ijms-24-13902-f003:**
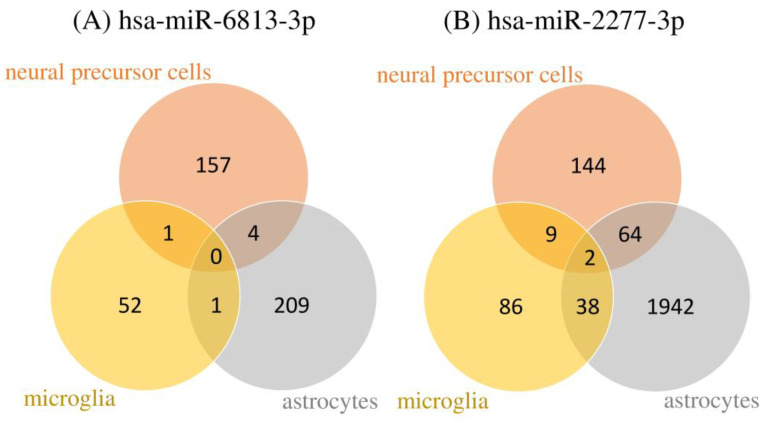
The Venn diagram shows the number of candidate target genes from microarray analysis of neural progenitor cells, astrocytes, and microglia.

**Figure 4 ijms-24-13902-f004:**
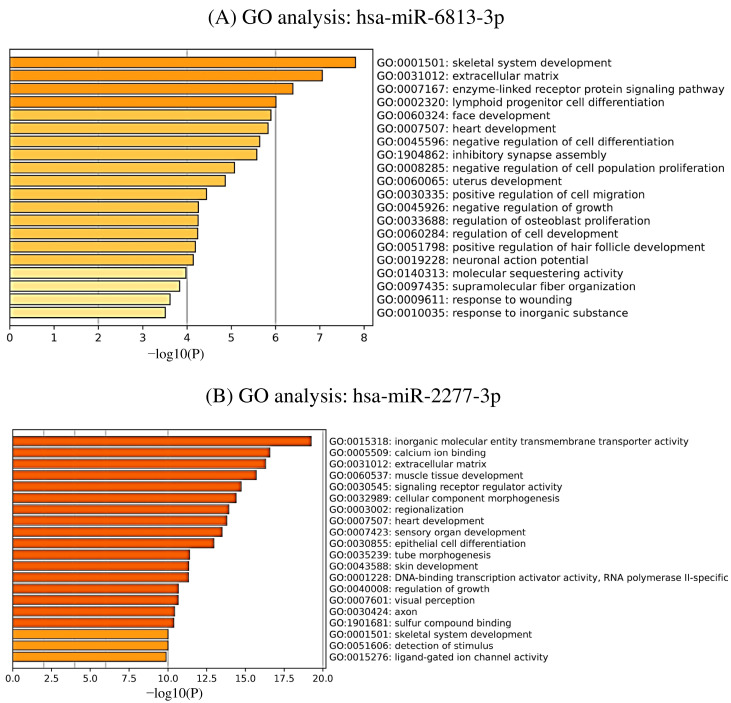
Functional analysis of candidate target genes and gene ontology (GO) enrichment analysis of candidate target genes in hsa-miR-6813-3p and hsa-miR-2277-3p. (**A**) GO analysis involving predicted target genes of hsa-miR-6813-3p. (**B**) Results of GO analysis involving predicted target genes of hsa-miR-6813-3p. *p* < 0.01 with the top 20 features listed.

**Table 1 ijms-24-13902-t001:** Demographic and clinical profile.

Variable	Value
HAMD score	16.50 ± 3.92
Age, (years)	50.13 ± 7.16
Sex	
Female, n (%)	8.0 (50.0)
Male, n (%)	8.0 (50.0)
Years of education	12.0 (1.50)
Ethnicity	
Japanese, n (%)	16 (100.0)
Drinking status	
Current drinker, n (%)	7.0 (43.75)
Lifetime abstainer, n (%)	9.0 (56.25)
Smoking status	
Current smoker, n (%)	2.0 (12.50)
Past smoker, n (%)	6.0 (37.50)
Never smoker, n (%)	8.0 (50.0)
Number of episode	1.0 (1.0)
Antidepressant use	
Yes, n (%)	12.0 (75.0)
No, n (%)	4.0 (25.0.)
Benzodiazepine use	
Yes, n (%)	10.0 (62.50)
No, n (%)	6.0 (37.50)
Care setting	
Inpatient, n (%)	14.0 (87.50)
Outpatient, n (%)	2.0 (12.50)

Normal distribution data is displayed as mean ± standard deviation. Non-normally distributed data is displayed as median (quartile deviation). Categorical variables are displayed as counts (percentages).

**Table 2 ijms-24-13902-t002:** Characteristics of depressed patients by severity of illness.

	Mild	Moderate	Severe	*p*-Value
HAMD score	12.0 (0.50)	16.0 (1.0)	23.0 (1.50)	0.003 *
Age, (years)	53.0 (3.75)	48.50 (3.88)	49.0 (6.75)	0.555
Sex				
Female, n (%)	2.0 (66.67)	6.0 (60.0)	0	0.354
Male, n (%)	1.0 (33.33)	4.0 (40.0)	3.0 (100.0)	
Years of education	16.0 (1.5)	12 (1.0)	12 (1.0)	0.304
Drinking status				
Current drinker, n (%)	2.0 (66.67)	5 (50.0)	0	0.377
Smoking status				
Current smoker, n (%)	0	1 (10.0)	1 (33.3)	0.376
Number of episode	3.0 (0.50)	1.0 (0.88)	0 (0.50)	0.194
Antidepressant use				
Yes, n (%)	3.0 (100.0)	7.0 (70.0)	2.0 (66.67)	0.777
No, n (%)	0	3.0 (30.0)	1.0 (33.33)	
Benzodiazepine use				
Yes, n (%)	1.0 (33.33)	7.0 (70.0)	2.0 (66.67)	0.764
No, n (%)	2.0 (66.67)	3.0 (30.0)	1.0 (33.33)	

Normal distribution data is displayed as mean ± standard deviation. Non-normally distributed data is displayed as median (quartile deviation). Categorical variables are displayed as counts (percentages). HAMD scores were categorized as follows: Mild (8–13), Moderate (14–18), and Severe (19 or more). *p*-values were compared in the three severity groups * *p* < 0.05 was considered statistically significant.

**Table 3 ijms-24-13902-t003:** Differentially expressed miRNAs associated with HAMD score severity in MD Patients.

miRNA	Control	Mild	Moderate	Severe	2-Group *p*-Value	2-Group FDR *p*-Value	4-Group *p*-Value	4-Group FDR *p*-Value
hsa-let-7f-1-3p	10.08	11.98	9.082	7.194	3.27 × 10^−1^	6.98 × 10^−1^	8.14 × 10^−2^	1.28 × 10^−1^
hsa-miR-2277-3p	11.34	7.750	6.410	0	6.63 × 10^−5^ *	1.07 × 10^−2^ *	6.25 × 10^−4^ *	3.94 × 10^−2^ *
hsa-miR-6813-3p	26.85	23.40	15.49	11.23	1.52 × 10^−5^ *	1.01 × 10^−3^ *	9.21 × 10^−5^ *	7.90 × 10^−3^ *

* *p* < 0.05 and decreasing expression with severity.

**Table 4 ijms-24-13902-t004:** Pathways associated with candidate target genes of hsa-miR-6813-3p (astrocyte control vs. astrocyte hsa-miR-6813-3p addition).

	Pathway	*p*-Value	Matched Entities	Pathway Entities
1	Burn wound healing	4.61 × 10^−4^	6	128
2	Dravet syndrome	7.57 × 10^−4^	3	20
3	GDNF-RET signaling axis	1.18 × 10^−3^	3	23
4	Differentiation Pathway	1.25 × 10^−3^	4	50
5	Pluripotent stem cell differentiation pathway	1.25 × 10^−3^	4	50
6	NRP1-triggered signaling pathways in pancreatic cancer	1.68 × 10^−3^	4	57
7	Development of ureteric collection system	2.35 × 10^−3^	4	61
8	Benzo(a)pyrene metabolism	2.48 × 10^−3^	2	9
9	Hair follicle development-organogenesis-part 2 of 3	2.94 × 10^−3^	3	32
10	Metapathway biotransformation Phase I and II	5.63 × 10^−3^	6	190
11	Oxidative Damage	6.20 × 10^−3^	3	40
12	Estrogen Receptor Pathway	6.69 × 10^−3^	2	13
13	Development of pulmonary dendritic cells and macrophage subsets	6.69 × 10^−3^	2	13
14	Hair Follicle Development-Cytodifferentiation (Part_3_of_3)	8.80 × 10^−3^	4	87
15	Nuclear Receptors Meta-Pathway	9.78 × 10^−3^	8	318
16	Estrogen metabolism	1.01 × 10^−2^	2	18
17	Tamoxifen metabolism	1.27 × 10^−2^	2	21
18	Cardiac Progenitor Differentiation	1.37 × 10^−2^	3	53
19	Complement activation	1.56 × 10^−2^	2	22
20	Complement system in neuronal development and plasticity	1.70 × 10^−2^	4	106
21	Imatinib and chronic myeloid leukemia	1.71 × 10^−2^	2	21
22	Complement and coagulation cascades	1.83 × 10^−2^	3	60
23	miRNA targets in ECM and membrane receptors	1.87 × 10^−2^	2	45
24	Hypothesized Pathways in Pathogenesis of Cardiovascular Disease	2.38 × 10^−2^	2	25
25	Angiotensin II receptor type 1 pathway	2.95 × 10^−2^	2	28
26	Glucocorticoid receptor pathway	2.99 × 10^−2^	3	71
27	Cannabinoid receptor signaling	3.15 × 10^−2^	2	31
28	Inflammatory response pathway	3.35 × 10^−2^	2	33
29	Endothelin Pathways	3.56 × 10^−2^	2	33
30	Amino acid conjugation of benzoic acid	3.79 × 10^−2^	1	4
31	Clock-controlled autophagy in bone metabolism	4.06 × 10^−2^	3	80
32	GABA receptor Signaling	4.22 × 10^−2^	2	37
33	Melatonin metabolism and effects	4.44 × 10^−2^	2	42
34	Sulindac Metabolic Pathway	4.71 × 10^−2^	1	6
35	IL-18 signaling pathway	4.75 × 10^−2^	6	287

**Table 5 ijms-24-13902-t005:** Pathways associated with candidate target genes of hsa-miR-2277-3p (astrocyte control vs astrocyte hsa-miR-2277-3p addition).

	Pathway	*p*-Value	Matched Entities	Pathway Entities
1	Metapathway biotransformation Phase I and II	9.95 × 10^−7^	37	190
2	GPCRs, Class A Rhodopsin-like	3.28 × 10^−4^	42	262
3	Development of ureteric collection system	9.31 × 10^−4^	14	61
4	Dopaminergic neurogenesis	1.13 × 10^−3^	9	30
5	Hair follicle development-organogenesis-part 2 of 3	1.48 × 10^−3^	9	32
6	Mammalian disorder of sexual development	1.70 × 10^−3^	8	25
7	Role of Osx and miRNAs in tooth development	1.79 × 10^−3^	6	37
8	Neural Crest Differentiation	2.20 × 10^−3^	19	101
9	Vitamin D Metabolism	2.26 × 10^−3^	5	11
10	Hair Follicle Development- Cytodifferentiation (Part_3_of_3)	2.29 × 10^−3^	17	87
11	TYROBP causal network in microglia	3.34 × 10^−3^	13	60
12	Oxidation by Cytochrome P450	3.34 × 10^−3^	13	64
13	Familial hyperlipidemia type 4	3.43 × 10^−3^	7	22
14	Nuclear receptors meta-pathway	4.24 × 10^−3^	44	318
15	1q21.1 copy number variation syndrome	7.49 × 10^−3^	7	39
16	Somatic sex determination	7.68 × 10^−3^	5	14
17	GPCRs, Other	9.34 × 10^−3^	17	114
18	Familial hyperlipidemia type 5	1.06 × 10^−2^	5	15
19	Thyroxine (Thyroid Hormone) Production	1.41 × 10^−2^	3	6
20	Estrogen metabolism	1.43 × 10^−2^	5	18
21	Prostaglandin and leukotriene metabolism in senescence	1.43 × 10^−2^	7	32
22	Oxidative Stress	1.43 × 10^−2^	7	29
23	Zinc homeostasis	1.57 × 10^−2^	8	37
24	Tryptophan metabolism	1.62 × 10^−2^	9	77
25	Pluripotent stem cell differentiation pathway	1.63 × 10^−2^	10	50
26	NRF2 pathway	1.66 × 10^−2^	21	143
27	Striated Muscle Contraction	1.85 × 10^−2^	8	38
28	Genes controlling nephrogenesis	1.88 × 10^−2^	9	44
29	Gene regulatory network modelling somitogenesis	2.22 × 10^−2^	4	12
30	Peptide GPCRs	2.26 × 10^−2^	13	75
31	Glial Cell Differentiation	2.29 × 10^−2^	3	8
32	Familial hyperlipidemia type 1	2.38 × 10^−2^	5	18
33	NO-cGMP-PKG mediated neuroprotection	2.49 × 10^−2^	9	47
34	White fat cell differentiation	2.91 × 10^−2^	7	32
35	Cell-type Dependent Selectivity of CCK2R Signaling	2.97 × 10^−2^	4	13
36	Development of pulmonary dendritic cells and macrophage subsets	2.97 × 10^−2^	4	13
37	MECP2 and Associated Rett Syndrome	3.21 × 10^−2^	11	90
38	Burn wound healing	3.31 × 10^−2^	16	128
39	Male infertility	3.50 × 10^−2^	20	145
40	FOXA2 pathway	3.67 × 10^−2^	5	21
41	Oligodendrocyte Specification and differentiation	3.92 × 10^−2^	6	31
42	Splicing factor NOVA regulated synaptic proteins	4.35 × 10^−2^	8	42
43	2q37 copy number variation syndrome	4.42 × 10^−2^	19	172
44	Methylation Pathways	4.75 × 10^−2^	3	9

## Data Availability

The data derived from subjects supporting the findings of this study cannot be opened due to privacy or ethical restrictions.

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
