# Peer review of "Serum Extracellular Vesicle-Derived hsa-miR-2277-3p and hsa-miR-6813-3p Are Potential Biomarkers for Major Depression: A Preliminary Study"

_ijms, 2023, doi:10.3390/ijms241813902_

Round 1

Reviewer 1 Report

The authors have proposed an interesting research topic - finding an objective and easily measurable marker of depression would be of special interest in the diagnose of depression and, in particular, in depression severity. Also, the subject of the study is very reasonable - extracellular vesicles have a great potential in miRNA research. The authors have found 3 possible miRNAs, which expression is decreased with depression severity, while 2 of them were different from control group. However, I have noticed a few considerable concerns in the methodology applied or at least in its description.

Major concerns:

1) Despite of a small size of patients, there is a good selection of them according to gender, smoking or drinking status. However, there I have found no demographic and clinical profile of the control group (healthy people), nor its size used in this study.

2) There is no explanation of the origin of the cells (i.e. neural progenitor cells, astrocytes and microglia) used in in vitro target screening  – were they a commercial cell lines? The authors declare 100% of efficiency of viral transfection of both plasmids in all cell lines, which was declared based on GFP expression. But was the expression of hsa-miR-6813-3p and hsa-miR-227-3p in the selected cell lines confirmed (e.g. by RT-PCR) to be definitely sure that the miRNAs was over-expressed? How many biological replicates were used in in vitro study? What was the control group (was it empty plasmid or scrambled miRNA)?

3) According to list of genes which expression was changed in specific cell lines (Supplementary Table 2), was the expression of these genes up-or down-regulated? It would be beneficial if the expression values of these genes were included in the supplementary Table 2. How many of listed genes were down-regulated and were a direct target of selected miRNAs (i.e. have a sequence for selected miRNAs in their 3’UTR, based on information available in miRNA databases)? Such information could clarify if the observed changes in gene expression resulted from direct interaction with specific miRNA or rather some secondary changes in the intracellular cascades.

4) GO and pathway enrichment analysis has revealed rather very general information about processes associated with target genes. Was the False Discovery Rate correction applied in the bioinformatics analyses? It would be beneficial for the readers if the authors list the genes associated to each term, at least in the supplementary table.

I have noticed minor text misspelling: In the table 1 it should be Antidepressant use, not “Antidependent use”.

To sum up, the authors are highly aware of the limitations of the presented study, what is mentioned in the last paragraph of the discussion. The size of the group of patients makes this study a preliminary. However, if the in vitro part of this study would be extended and methodology would be better described, it would gain far more scientific value.

Author Response

For Reviewer 1

Major concerns:

1) Despite of a small size of patients, there is a good selection of them according to gender, smoking or drinking status. However, there I have found no demographic and clinical profile of the control group (healthy people), nor its size used in this study.

Answer;

Thank you for pointing out the need for more details about the control group. We did mention in the "4.1 Subjects" section that the control group included 284 healthy individuals aged 20–68 with no specific diseases. However, we recognize your request for more information. In response, we've added details on gender distribution, mean age, and other demographics in both the "Patient Characteristics" and "4.1 Subjects" sections. Unfortunately, specific information on smoking habits, alcohol consumption, and education was not available.

2) There is no explanation of the origin of the cells (i.e. neural progenitor cells, astrocytes and microglia) used in in vitro target screening – were they a commercial cell lines? The authors declare 100% of efficiency of viral transfection of both plasmids in all cell lines, which was declared based on GFP expression. But was the expression of hsa-miR-6813-3p and hsa-miR-227-3p in the selected cell lines confirmed (e.g. by RT-PCR) to be definitely sure that the miRNAs was over-expressed? How many biological replicates were used in in vitro study? What was the control group (was it empty plasmid or scrambled miRNA)?

Answer;

Thank you for pointing out. We forgot to mention the cell origins. All cells used in this study were purchased. This information is described in 4.5 section. Expression analysis of hsa-miR-2277-3p and hsa-miR-6813-3p was performed by RT-qPCR using stored RNA. Supplementary Tables 6 and 7 show that hsa-miR-6813-3p and hsa-miR-2277-3p are overexpressed in stable transfectants. Briefly, we confirmed that the three cell lines of hsa-miR-2277-3p-transfected cells expressed more than 10-fold. On the other hand, miR-6813-3p was not amplified with the control empty vector, but the Ct value was confirmed in the three cell lines of miR-6813-3p-transfected cells, suggesting that it is highly expressed. In vitro analysis is a single analysis. A control group used a vector without an insert. We added this explanation to 4.5 section.

3) According to list of genes which expression was changed in specific cell lines (Supplementary Table 2), was the expression of these genes up-or down-regulated? It would be beneficial if the expression values of these genes were included in the supplementary Table 2. How many of listed genes were down-regulated and were a direct target of selected miRNAs (i.e. have a sequence for selected miRNAs in their 3’UTR, based on information available in miRNA databases)? Such information could clarify if the observed changes in gene expression resulted from direct interaction with specific miRNA or rather some secondary changes in the intracellular cascades.

Answer;

Thank you for pointing out. Since miRNA overexpression aims to identify the direct target mRNAs of miRNAs, only mRNAs with decreased expression are listed. TargetScan predicts biological targets of miRNAs by searching for the presence of conserved 8mer, 7mer, and 6mer sites that match the seed region of each miRNA (https://www.targetscan.org/vert_80/). Supplementary Table 2 shows genes found in common between our results and TargetScan. Thus, the genes listed in Supplementary Table 2 may be the direct targets of each miRNA in each cell. On the other hand, the unlisted genes may be secondarily repressed. We do not plan to perform further analyses, as we do not consider these analyzes to be the main focus of this study.

4) GO and pathway enrichment analysis has revealed rather very general information about processes associated with target genes. Was the False Discovery Rate correction applied in the bioinformatics analyses? It would be beneficial for the readers if the authors list the genes associated to each term, at least in the supplementary table.

Answer;

Thank you for pointing out and commenting. For the analysis conducted through Metascape, the FDR (False Discovery Rate) correction was performed using the Benjamini-Hochberg method. The specifics are described as follows: "In Metascape, FDR correction was conducted using the Benjamini-Hochberg method, and adjusted p-values were employed [53]." Concerning pathway analysis, FDR correction was not performed because "group comparison" was not performed in this study. In other words, this analysis aims to evaluate whether the set of genes (differentially expressed genes) is related to specific biological processes or pathways. In the context of our research, there was no need for FDR correction. Regarding the inclusion of a list of genes associated with each term in the supplementary table, we regret to inform you that the analysis was conducted using GeneSpringGX (Agilent), and due to a version change, we are unable to generate the gene list at this time.

Other author’s comment

I have noticed minor text misspelling: In the table 1 it should be Antidepressant use, not “Antidependent use”.

Answer;

Thank you for comment. In Table 1, the term "Antidepressant use" was incorrectly listed as "Antidependent use" when it should have been "Antidepressant use". This error has been corrected in the revised manuscript.

Reviewer 2 Report

In this manuscript, Seki et al explored a relationship between the severity of Major Depression (MD) and the levels of miRNAs found in extracellular vesicles (EVs) from peripheral blood serum. Because we still lack objective biomarkers for the severity of MD, the topic is of great interest. However, the manuscript has major weaknesses to allow proper interpretation of the results.

Major concerns:

-Surprisingly, the authors did not mention the work of Wei et al on exosomes from patients with MD (PMID: 31986519). To our knowledge, this study was the first to explore the miRNA expression profile of serum exosome in MD patients and healthy control subjects to identify potential MD markers by microRNA sequencing. Nor did the authors cite Ran et al’s work of on serum EV microRNA dysregulation and childhood trauma in adolescents with MD (PMID: 35659238). Most importantly, the authors neglected to cite the work of Honorato-Mauer et al on alterations in microRNA of EVs associated with MD, attention-deficit/hyperactivity and anxiety disorders in adolescents (PMID: 36746925). Overall, few studies to date have examined EVs in neuropsychiatric disorders, and the present evidence appears to lack reproducibility. It is therefore important to compare the results obtained in the present manuscript compared with previous findings and discuss potential discrepancies. The authors may wish to consult the review of Kohshour et al (PMID: 36302978).

- The main limitation of the current work, as acknowledged at the end of the Discussion is the very small sample size. Most patients suffer from moderate depression, while only three male patients are classified as severely depressed (no women in the severely depressed category, which is a bit of a surprise recruitment...). Consequently, the cohort does not present a true gradient of depression severity that reflects both prevalence in the general population and sex ratio imbalance.  The heterogeneity of age, number of previous episodes and antidepressant treatment makes it impossible, with a sample of this size, to draw any solid interpretation of the correlation between a specific miRNA profile and the severity of depression.

-Lines 316-318: the way in which the sera were obtained is unconventional. Blood is supposed to clot before centrifugation and serum collection. Please provide further details on the serum collection procedure.

- What do the values in Table 3 mean? It's puzzling that so many groups have a "0" value, while the next small value jumps to 4.51. The authors should explain further how they quantified miRNA expression levels and what kind of filtering procedures, if any, they applied to avoid background noise. It also appears that none of the statistical values are corrected for multiple testing. We are concerned that, due to the 2565 probe set used, none of the miRNAs may still be significant after FDR or Bonferroni correction.

- To associate candidate miRNA expression with depression severity, the authors could have performed a regression analysis. Why, at least, didn't they generate Spearman correlations?

-Lines 111-117. The authors sought to confirm the downregulation of the expression of 3 candidate miRNAs as a function of depression severity, by comparing this expression with that of healthy controls. Nowhere did we find the technique applied to test miR-6813-3p, miR-2277-3p and let-7f-1-3p in both controls and MD patients. Furthermore, did the authors really test 284 control sera? This is really intriguing because the values for MD patients in Figure 1 correspond to the values in Table 3, so we assume that the values for controls were obtained after microarray profiling. But doesn't that make sense?

- From paragraph 2.3 onwards, the reader is completely lost. Where do these neuron precursor cells, astrocytes and microglia come from? The authors should try to write a story where the reader is not supposed to guess what they wanted to explain by looking at the methods. It's impossible for the reader to evaluate exactly what was done and relate it to previous results in the manuscript. Furthermore, the previous result was a down-regulation of the expression of 2 miRNAs, and the authors then over-expressed these miRNAs: this doesn't make sense, it should be the other way around. The authors should explain why he sought to study different brain cell types in which he overexpressed the candidate miRNAs. In addition, the authors did not provide any proof that transfected cells overexpress the 2 candidate miRNAs.

- Unless the authors try to explain what they have achieved with the transfected cells, with the number of biological replicates or technical replicates for microarray profiling, all GO analysis is purely speculative and cannot be appreciated by the reviewer.

- If the authors could prove that the main EVs miRNA signal originates from astrocytes, then all the work with transfected cells might make sense. If not, this work has no connection with the first part of the study and only makes it very confusing.

Minor concerns:

-In the Title, line 2, “TSerum” should be replaced by “Serum”

-In the Abstract, line 23, we should read “The predicted target genes underwent enrichment analysis”.

-In the Abstract, line 25-26: too much “pathway” wording. The sentence could be simplified: “Pathway enrichment analysis suggested that has-miR-6813-3p may be involved in glucocorticoid receptor and gamma-aminobutyric acid receptor signaling”.

- Given that miRNAs in major depression are the focus of the manuscript, the authors could cite in the introduction a more recent and comprehensive review by a group leader in this field. I therefore suggest adding or replacing reference 11 (line47) with the work by Zurawek and Turecki (“The miRNome of depression”, PMID: 34768740) also published in the International Journal of Molecular Sciences.

-In the Introduction, lines 56-58, we are not sure that ref 18 is adequate and suggest for example to replace it with the work of Gao et al (“A New Player in Depression: MiRNAs as Modulators of Altered Synaptic Plasticity, PMID: 35562946).

-Line79: 3 had severe or severe depression?

-Lines 316-318: the way in which the sera were obtained is unconventional. Blood is supposed to clot before centrifugation and serum collection. Please provide further details on the serum collection procedure.

-Lines 105-106: The sentence is not at the right place: it sounds like a duplication of the title of the paragraph?

-Table 3: please use a maximum of 4 significant numbers and adopt a scientific format for p-values

-Lines 129-141: All this part is redundant with lines 111-117

-Supplementary Tables were not provided to the reviewer.

Apart from a few clumsinesses, the quality of the English is sufficient to understand the text.

Author Response

For Reviewer 2

Major concerns:

1) Surprisingly, the authors did not mention the work of Wei et al on exosomes from patients with MD (PMID: 31986519). To our knowledge, this study was the first to explore the miRNA expression profile of serum exosome in MD patients and healthy control subjects to identify potential MD markers by microRNA sequencing. Nor did the authors cite Ran et al’s work of on serum EV microRNA dysregulation and childhood trauma in adolescents with MD (PMID: 35659238). Most importantly, the authors neglected to cite the work of Honorato-Mauer et al on alterations in microRNA of EVs associated with MD, attention-deficit/hyperactivity and anxiety disorders in adolescents (PMID: 36746925). Overall, few studies to date have examined EVs in neuropsychiatric disorders, and the present evidence appears to lack reproducibility. It is therefore important to compare the results obtained in the present manuscript compared with previous findings and discuss potential discrepancies. The authors may wish to consult the review of Kohshour et al (PMID: 36302978).

Answer;

Thank you for pointing out the lack of discussion of relevant studies on exosomes or microRNAs in MD patients. We added references reviewed by Wei et al., Ran et al., Honorato-Mauer et al., Oraki Kohshour et al. Furthermore, we discussed the differences between the miRNAs we have identified and previous reports. In particular, we described that the reasons for the discrepancy in the results are thought to be that there is no established method for isolating only exosomes, and that microvesicles are somewhat contaminated in individual isolation methods.

2) The main limitation of the current work, as acknowledged at the end of the Discussion is the very small sample size. Most patients suffer from moderate depression, while only three male patients are classified as severely depressed (no women in the severely depressed category, which is a bit of a surprise recruitment...). Consequently, the cohort does not present a true gradient of depression severity that reflects both prevalence in the general population and sex ratio imbalance.  The heterogeneity of age, number of previous episodes and antidepressant treatment makes it impossible, with a sample of this size, to draw any solid interpretation of the correlation between a specific miRNA profile and the severity of depression.

Answer;

Thank you for pointing out the limitations in our sample size and the representation of depression severity. We agree that these issues affect the interpretation of our results. As noted in the "Limitations" section, we plan to address these challenges in future research by employing a larger and more diverse sample. Your insights are valuable and will guide our next steps.

3) Lines 316-318: the way in which the sera were obtained is unconventional. Blood is supposed to clot before centrifugation and serum collection. Please provide further details on the serum collection procedure.

Answer;

As you point out, we had to explain how the serum was prepared. In the section 4.3, we added the method of preparing the serum in this experiment.

4) What do the values in Table 3 mean? It's puzzling that so many groups have a "0" value, while the next small value jumps to 4.51. The authors should explain further how they quantified miRNA expression levels and what kind of filtering procedures, if any, they applied to avoid background noise. It also appears that none of the statistical values are corrected for multiple testing. We are concerned that, due to the 2565 probe set used, none of the miRNAs may still be significant after FDR or Bonferroni correction.

Answer;

Thank you for pointing out the statistical analysis. We were able to do the statistical analysis correctly. We have updated and organized the content clearly, separating it between Table 3 and the Supplementary Table. Only 3 miRNAs with significant differences were included in Table 3. In addition, although the results of depression patients were presented, the results of healthy subjects were also presented. Multiple negative controls are prepared as probes for the microarray. The values in Table 3 represent the miRNA signal values adjusted by subtracting the mean value ± 2SD of these negative controls. MiRNAs with an adjusted value of zero or less are shown as "0," indicating that the specific expression of that miRNA was not detected. The jump from "0" to 4.51 reflects the situation where, during data preprocessing to remove noise, miRNAs with very low expression levels were all set to "0," and the next validly detected miRNA signal value was 4.51. This explanation has been added to the notes section of Table 3 for further clarity. However, during our data reorganization, the specific value "4.51" you referenced was removed from Table 3. Thank you for pointing out the FDR correction, which is an important issue in multiplicity tests. New FDR-corrected p-values were obtained using JMP's statistical analysis software. JMP Response Screening uses multiple adjusted p-values for FDR (False Discovery Rate) as described in Benjamini and Hochberg (1995). We have added these results to Table 3.

5) To associate candidate miRNA expression with depression severity, the authors could have performed a regression analysis. Why, at least, didn't they generate Spearman correlations?

Answer;

Thank you for your suggestion. We have included Spearman's correlation in the revised manuscript, as presented in Figure 2.

6) Lines 111-117. The authors sought to confirm the downregulation of the expression of 3 candidate miRNAs as a function of depression severity, by comparing this expression with that of healthy controls. Nowhere did we find the technique applied to test miR-6813-3p, miR-2277-3p and let-7f-1-3p in both controls and MD patients. Furthermore, did the authors really test 284 control sera? This is really intriguing because the values for MD patients in Figure 1 correspond to the values in Table 3, so we assume that the values for controls were obtained after microarray profiling. But doesn't that make sense?

Answer;

Extracellular vesicle isolation and miRNA microarray analysis from sera obtained from healthy subjects (284) and MD patients (16) were performed consistently. However, the "4.3 Laboratory methods" section was revised because the methods for separating and storing serum were not sufficiently explained. Isoloation of EVs and Microarray analysis were described at 4.4 and 4.6 section, respectively. Through these series of analyses, we identified has-miR-6813-3p, has-miR-2277-3p and let-7f-1-3p. The data presented in Figure 1 are a direct result of this consistent methodology.

7) From paragraph 2.3 onwards, the reader is completely lost. Where do these neuron precursor cells, astrocytes and microglia come from? The authors should try to write a story where the reader is not supposed to guess what they wanted to explain by looking at the methods. It's impossible for the reader to evaluate exactly what was done and relate it to previous results in the manuscript. Furthermore, the previous result was a down-regulation of the expression of 2 miRNAs, and the authors then over-expressed these miRNAs: this doesn't make sense, it should be the other way around. The authors should explain why he sought to study different brain cell types in which he overexpressed the candidate miRNAs. In addition, the authors did not provide any proof that transfected cells overexpress the 2 candidate miRNAs.

Answer;

Thank you for your comments. We did not adequately explain the meaning of overexpressing miRNAs. First, the expression levels of the two miRNAs are low in the cell lines used in this assay. Although the cells that secrete EVs containing the two miRNAs are unknown, our results show that their expression is lower in serum EVs from MD patients than in healthy subjects. Thus, we hypothesized that cells involved in neuronal function (mimicked in cultured cells) have low expression of these two miRNAs but can uptake these miRNAs from serum EVs and exert their normal functions. Therefore, we searched for a mechanism that positively affects neuronal function by expressing these two miRNAs in cultured cells with low expression levels. This is the interpretation that the function can be maintained by suppressing the expression of the suppressing gene). We explained these explanations in the first paragraph of 2.3.

8) Unless the authors try to explain what they have achieved with the transfected cells, with the number of biological replicates or technical replicates for microarray profiling, all GO analysis is purely speculative and cannot be appreciated by the reviewer.

Answer;

Thank you for your comments. This answer is included in the previous explanation.

9) If the authors could prove that the main EVs miRNA signal originates from astrocytes, then all the work with transfected cells might make sense. If not, this work has no connection with the first part of the study and only makes it very confusing.

Answer;

Thank you for your comments. Currently, there is no way to determine which cells secreted EVs in the blood. In the present analysis, the expression of has-miR-6813-3p and has-miR-2277-3p in astrocyte cells is low, we do not believe that astrocytes supply these miRNAs using EVs.

Minor concerns:

  • In the Title, line 2, “TSerum” should be replaced by “Serum”

Answer;

We've corrected the typo in the title from "TSerum" to "Serum".

2) In the Abstract, line 23, we should read “The predicted target genes underwent enrichment analysis”.

Answer;

We have revised the abstract to reflect the statement "The predicted target genes underwent enrichment analysis."

3) In the Abstract, line 25-26: too much “pathway” wording. The sentence could be simplified: “Pathway enrichment analysis suggested that has-miR-6813-3p may be involved in glucocorticoid receptor and gamma-aminobutyric acid receptor signaling”.

Answer;

In the abstract, we have now simplified the sentence to: “Pathway enrichment analysis suggested that has-miR-6813-3p may be involved in glucocorticoid receptor and gamma-aminobutyric acid receptor signaling.”

4) Given that miRNAs in major depression are the focus of the manuscript, the authors could cite in the introduction a more recent and comprehensive review by a group leader in this field. I therefore suggest adding or replacing reference 11 (line47) with the work by Zurawek and Turecki (“The miRNome of depression”, PMID: 34768740) also published in the International Journal of Molecular Sciences.

Answer;

We have incorporated the work by Zurawek and Turecki, "The miRNome of depression", into reference 11 (line 47) for a more comprehensive understanding of miRNAs in major depression.

5) In the Introduction, lines 56-58, we are not sure that ref 18 is adequate and suggest for example to replace it with the work of Gao et al (“A New Player in Depression: MiRNAs as Modulators of Altered Synaptic Plasticity, PMID: 35562946).

Answer;

We have replaced reference 18 with the work of Gao et al., "A New Player in Depression.

6) Line79: 3 had severe or severe depression?

Answer;

We have corrected the phrasing in line 79 to avoid repetition.

7) Lines 316-318: the way in which the sera were obtained is unconventional. Blood is supposed to clot before centrifugation and serum collection. Please provide further details on the serum collection procedure.

Answer;

After blood collection, it was allowed to clot at room temperature by slowly rotating for 5 minutes. Once coagulated, it was centrifuged at 3,000 rpm for 10 minutes, and the supernatant serum was stored at -80°C.

8) Lines 105-106: The sentence is not at the right place: it sounds like a duplication of the title of the paragraph?

Answer;

We've removed the redundant sentence from lines 105-106 to avoid duplication with the title of the paragraph.

9) Table 3: please use a maximum of 4 significant numbers and adopt a scientific format for p-values

Answer;

In Table 3, we've adjusted the values to use a maximum of four significant figures and have adopted a scientific format for p-values.

10) Lines 129-141: All this part is redundant with lines 111-117

Answer;

We've removed the redundant information from lines 129-141 to maintain clarity and avoid repetition.

11) Supplementary Tables were not provided to the reviewer.

Answer;

We apologize for the oversight. We have now provided the Supplementary Tables for your review.    

Round 2

Reviewer 1 Report

The authors addressed my main concerns and improved the description of methodological issues in the manuscript and in the supplementary files. However some of the methodological aspects  were not clearly depicted and should be clearly written in the manuscript:

1)      The authors wrote that control group consisted of 248 individuals. Were all the patients in the control group (i.e. 248 cases) analyzed separately? Or  control sample was pooled from these individuals and miRNA was then analyzed? It makes the difference and should be clearly indicated in the manuscript.

2)      The number of biological replicates in cell experiments is still missing – from supplementary file I guess it was 9, but it should be clearly indicated in the text.

3)      table 3 is missing in the manuscript.

4)      please indicate the catalog numbers of cell lines.

To sum up, it is a preliminary study with very low number of patients, albeit the idea of looking for objective and easily measurable biomarkers  of depression severity is interesting. The authors have found 2 possible miRNAs and checked their possible role in cell lines derived from brain. The GO and pathway enrichment analysis has revealed rather very general information about processes associated with targets of these miRNAs, but it is not surprising in such analyses. Overall, I find this paper worth attention.

Author Response

For Reviewer 1

Q1: The authors wrote that control group consisted of 248 individuals. Were all the patients in the control group (i.e. 248 cases) analyzed separately? Or  control sample was pooled from these individuals and miRNA was then analyzed? It makes the difference and should be clearly indicated in the manuscript.

A1: Thank you for your comment. We collected a total of 300 sera for miRNA array analysis in the control group. All 300 samples were analyzed without pooling. However, 284 samples were included in the final dataset, because 16 samples were later excluded due to defects in either the samples themselves or the array chips. We have revised the "Microarray analysis" section of our manuscript to incorporate this information more clearly.

Q2: The number of biological replicates in cell experiments is still missing – from supplementary file I guess it was 9, but it should be clearly indicated in the text.

A2: Thank you for your careful review and comments on our manuscript. Regarding your comment on the number of biological replicates in cell experiments: We apologize for the oversight in our previous versions. We have now clarified this in the revised manuscript. As you rightly pointed out, the number of biological replicates was not explicitly mentioned in the main text. In the section "Formation of cells stably expressing miRNA and purification of total RNA", we have now added the statement, "Cells stably expressing miRNAs were established using lentivirus with a biological replicate of 1." Also, We recognize that the presentation of Table S3 was not clear and lacked the necessary sample information. To address this, we have revised Table S3 to not only clarify the singlicate nature of the experiments but also to include the correct sample names.

Q3: table 3 is missing in the manuscript.

A3: Thank you for your feedback. We've revised the manuscript for better clarity and coherence. The mentioned "Table 3" was an oversight and should have referred to "TableS1". This has been corrected. To emphasize the importance, we've highlighted the title of Table 3 in red.

Q4: please indicate the catalog numbers of cell lines.

A4: Thank you for your comment. We added the catalog number of each cell line in the text.

Reviewer 2 Report

We thank the authors for having thoroughly revised their manuscript, taking into account most of the concerns and recommendations expressed by the reviewer. We would nevertheless suggest improving the readability of Figure 1 by somehow enlarging the x- and y-axis text, as it is impossible to read in the current version. The same applies to figure 2 (although it's roughly legible when enlarged in its current form).

More importantly, we regret that the hypothesis presented in the revised manuscript from line 162 to 165 lacks credibility. Indeed, the authors suggest that peripheral information circulating in the blood would cross the blood brain barrier and feed neuroglial cells. This is interesting, but without any scientific basis provided by the authors. In its present form, the entire second part of the article is based on this unjustified hypothesis. As current discussion in the revised manuscript focuses on the reverse mechanism (information flows from the brain to the periphery), we maintain our view that the authors should either carry out a further experiment or provide bibliographical references to support the scientific validity of their assertion. As an alternative approach, the authors could try to better synthesize the wealth of information provided by the GO analysis to prove that overexpression of the candidate miRNA "feeds" neuroglial cells and maintains their fitness. Furthermore, there is no method section for the TargetScan approach.

In supplementary Tables 2-4, it is "Microarray" and not "Macroarray"

Lines 179-180: "PCDHA4" is repeated four times in a row?

Line 253: "suspended" and not "sus-pended"

Line 255: "comprehensive" and not "com-prehensive"

Author Response

For Reviewer 2

Q1: We thank the authors for having thoroughly revised their manuscript, taking into account most of the concerns and recommendations expressed by the reviewer. We would nevertheless suggest improving the readability of Figure 1 by somehow enlarging the x- and y-axis text, as it is impossible to read in the current version. The same applies to figure 2 (although it's roughly legible when enlarged in its current form).

A1: Thank you for your valuable feedback and suggestions. In response to your comments regarding the readability of the x- and y-axis text in Figure 1 and Figure 2, we have adjusted the size of the text in these figures. Additionally, we took the liberty to enhance the readability of other figures and tables in our manuscript. We believe this adjustment enhances the readability of both figures and other included visuals, and hope that it now meets your expectations. Once again, we appreciate your constructive feedback which greatly aids in improving the quality of our manuscript.

Q2: More importantly, we regret that the hypothesis presented in the revised manuscript from line 162 to 165 lacks credibility. Indeed, the authors suggest that peripheral information circulating in the blood would cross the blood brain barrier and feed neuroglial cells. This is interesting, but without any scientific basis provided by the authors. In its present form, the entire second part of the article is based on this unjustified hypothesis. As current discussion in the revised manuscript focuses on the reverse mechanism (information flows from the brain to the periphery), we maintain our view that the authors should either carry out a further experiment or provide bibliographical references to support the scientific validity of their assertion. As an alternative approach, the authors could try to better synthesize the wealth of information provided by the GO analysis to prove that overexpression of the candidate miRNA "feeds" neuroglial cells and maintains their fitness. Furthermore, there is no method section for the TargetScan approach.

A1: Thank you for your valuable comments and feedback on our manuscript. Regarding your concern about the credibility of our hypothesis presented in lines 162 to 165, we have taken your feedback into account and enhanced the "Discussion" section of our manuscript to incorporate supporting evidence from prior research. We have now provided references and detailed insights from studies that have established the capability of extracellular vesicles, especially exosomes, to pass through the blood-brain barrier in both directions. In particular, the study [PMID: 36678926] deeply investigates the mechanism through which exosomes enter brain tissues. Several other studies, such as [PMID: 33717653], [PMID: 28970251], [PMID: 35308117], and [PMID: 29883579], have reported the observed neuroregenerative effects following the intravenous injection of exosomes into mice. Additionally, the research by Chen et al. [PMID: 34073900] demonstrated the positive impacts of mesenchymal stem cells (MSC)-derived exosomes on the functioning of astrocytes in an Alzheimer's disease model. We believe that the inclusion of these references and expanded discussion will address your concerns regarding the foundation of our hypothesis. We hope this enhancement strengthens our argument and provides a clearer scientific basis for our claims. We've added a "4.8 Selection of candidate target genes" section detailing our use of the TargetScan™ database for miRNA target prediction.

Q3: In supplementary Tables 2-4, it is "Microarray" and not "Macroarray"

A3: Thank you for your careful review and for pointing out the error in supplementary Tables. We have corrected the term "Macroarray" to "Microarray" as suggested.

Q4: Lines 179-180: "PCDHA4" is repeated four times in a row?

A4: Thank you for pointing out the repetition. We have corrected the error in lines 179-180 by removing the repeated mentions of "PCDHA4".

Q5: Line 253: "suspended" and not "sus-pended"

A5: Thank you for pointing out the typographical error. We have corrected line 253 from "sus-pended" to "suspended". However, please note that due to an error in our editing program, hyphens are sometimes automatically inserted in multiple places when copying and pasting. We are being careful to address this issue. Thank you for bringing it to our attention.

Q6: Line 255: "comprehensive" and not "com-prehensive"

A6: Thank you for catching the typographical error. We have corrected line 255, changing "com-prehensive" to "comprehensive."

Round 3

Reviewer 2 Report

We thank the authors for taking account of all our additional remarks. We hope the authors will follow with enlarged sample size in the future to provide validation of their preliminary results.